# Offloading under cognitive load: Humans are willing to offload parts of an attentionally demanding task to an algorithm

**Basil Wahn**[1,2]*, Laura Schmitz[3], Frauke Nora Gerster[1], Matthias Weiss[4]

**1** Institute of Educational Research, Ruhr University Bochum, Bochum, Germany, **2** Department of Neurophysiology and Pathophysiology, University Medical Center Hamburg-Eppendorf, Hamburg, Germany, **3** Department of Neurology, University Medical Center Hamburg-Eppendorf, Hamburg, Germany, **4** Nijmegen School of Management, Radboud University Nijmegen, Nijmegen, The Netherlands

* basil.wahn@rub.de

**Data Availability Statement:** Our full data set and the analysis script are available on OSF: https://osf.io/qbgm3/

## Abstract

In the near future, humans will increasingly be required to offload tasks to artificial systems to facilitate daily as well as professional activities. Yet, research has shown that humans are often averse to offloading tasks to algorithms (so-called "algorithmic aversion"). In the present study, we asked whether this aversion is also present when humans act under high cognitive load. Participants performed an attentionally demanding task (a multiple object tracking (MOT) task), which required them to track a subset of moving targets among distractors on a computer screen. Participants first performed the MOT task alone (Solo condition) and were then given the option to offload an unlimited number of targets to a computer partner (Joint condition). We found that participants significantly offloaded some (but not all) targets to the computer partner, thereby improving their individual tracking accuracy (Experiment 1). A similar tendency for offloading was observed when participants were informed beforehand that the computer partner's tracking accuracy was flawless (Experiment 2). The present findings show that humans are willing to (partially) offload task demands to an algorithm to reduce their own cognitive load. We suggest that the cognitive load of a task is an important factor to consider when evaluating human tendencies for offloading cognition onto artificial systems.

## Introduction

Cognitive offloading is defined as "the use of physical action to alter the information processing requirements of a task so as to reduce cognitive demand" [1, p. 676]. As such, offloading can help humans to overcome cognitive capacity limitations [2, 3], enabling them to attain goals that could not have been attained (as quickly, easily, or efficiently) otherwise. In the present paper, we specifically focus on offloading cognition onto technological devices [1]. For example, consider using *Google Maps* (or a similar navigation app on your phone) while driving, to guide you to your destination–by doing so, you free up the cognitive capacity normally required for paying attention to road signs, which in turn allows you to focus entirely on the

**Funding:** This study is part of the cooperative research project "INTERACT!", which received funding from the program "Profilbildung 2020" — an initiative of the Ministry of Culture and Science of the State of North Rhine-Westphalia. The sole responsibility for the content of this publication lies with the authors. The funders had no role in study design, data collection and analysis, decision to publish, or preparation of the manuscript.

**Competing interests:** The authors have no competing interests to declare.

busy traffic or even to perform a secondary task such as talking to your friend in the passenger seat. Or consider using virtual assistants such as *Siri* or *Alexa* to retrieve information that you would otherwise spend time and cognitive effort looking up manually (e.g., by searching for it on the internet). In the near future, humans will likely (be required to) interact more frequently with increasingly more sophisticated artificial systems (e.g., *ChatGPT*) to facilitate daily as well as professional activities [4]. For this reason, it is both necessary and timely to further investigate the conditions under which humans are willing to offload certain tasks to artificial systems.

Recent research [5–8] on cognitive offloading has investigated a number of tasks and factors that influence the human likelihood to offload cognition to technological devices. For instance, one study [5] used a mental rotation task in which participants could reduce their cognitive demand by externalizing the task by manipulating a knob to physically rotate the stimulus. The (actual and believed) reliability of that knob was systematically varied. It was found that participants adapted their offloading behavior such that they used the knob to a smaller extent if its reliability was (believed to be) lower. This finding suggests that preexisting beliefs about the technological reliability, even if false, can influence offloading behavior. In a follow-up study [6], researchers instructed participants to either focus on the speed or the accuracy of their performance. If the focus was on accuracy, participants offloaded more frequently compared to if the focus was on speed, suggesting that offloading behavior is influenced by performance goals. In a different study [7], participants were required to solve an arithmetic or a social task with the aid of either another human, a (picture of a) robot, or an app. These potential "assistants" were described as having either task-specific or task-unspecific expertise. It was found that the different descriptions greatly influenced offloading behavior with regard to the app (i.e., more offloading for task-specific vs. task-unspecific expertise) but less so with regard to the human and robot. This suggests that the expertise of a technological system is another factor that influences offloading behavior in humans. It was also found that participants offloaded the arithmetic task to a "socially incapable" robot while offloading the social task to a "socially capable" robot even though both robots were equally capable of solving the arithmetic task [8]. This suggests that humans tend to implicitly assume that artificial systems are designed to be skilled in only one task domain (e.g., the social domain), but not in two different domains (e.g., social and arithmetic). In sum, the above-mentioned studies show that the human willingness to offload tasks to technological systems is influenced by the system's (actual and believed) reliability [5], its (ascribed) expertise [7, 8], as well as participants' own performance goals [6].

Other studies compared the human willingness to offload tasks to another human vs. a technological system. In particular, these studies identified several factors that influence whether humans tend to show "algorithmic appreciation" or "algorithmic aversion", i.e., whether humans prefer to rely on an assessment made by an algorithm rather than by a human, or the other way around (for a recent review, see [4]). Algorithmic appreciation/aversion has often been tested in decision-making contexts. In these studies, an algorithm either aids participants in the decision-making process by taking an advisory role [9] or participants are asked whether a particular decision should be taken by another human or by an algorithm [10]. For instance, in the latter case, participants were asked in one study [10] whether it is more permissible that a medical decision (i.e., performing a risky surgery or not) is made by a human doctor or by an algorithm. Participants clearly preferred the human doctor, suggesting algorithmic aversion. However, if participants were told that the expertise of the algorithm was higher than that of the human doctor, participants' preference shifted to algorithmic appreciation. Note that medical decision-making often involves particularly high stakes, since making a wrong decision can have serious consequences. In contrast, many everyday tasks that

humans might potentially offload to algorithms involve comparatively low stakes, such as asking *Siri* for advice or using *Google Maps* to plan a route. Even if the respective algorithm failed (e.g., *Siri* does not provide the desired information or *Google Maps* does not plan the most efficient route), the consequences for the human user would normally not be serious (e.g., spending some extra time to retrieve the desired information or to plan the route).

A related set of studies investigated situations where algorithms can perform tasks autonomously–so-called "performative algorithms" [4]. For instance, one study tested whether humans prefer to rely on another human's forecast or on the forecast of an algorithm [11]. These forecasts predicted students' academic success based on their admission data. Participants tended to rely on the human forecaster, even if the algorithm made more accurate predictions than the human. Another study [12] investigated how humans judge the authenticity of works (e.g., art or music) created by either humans or algorithms. Participants perceived the algorithmic work as less authentic and genuine than the human work. Yet, if participants were told that an algorithm had created the work while being guided by a human, they judged this work as more authentic (than if an algorithm had created it autonomously).

To summarize (see [4]), key factors that determine whether humans show algorithmic appreciation or aversion are whether the algorithm (1) performs tasks on its own [12] or has an advisory role [9], (2) makes mistakes or performs flawlessly [11], and (3) is perceived as capable of performing the assigned task or not [10]. With regard to the first point, people are more likely to feel aversion towards an algorithm that performs a task autonomously compared to an algorithm in an advisory role [4]. With regard to the second point, if people observe an algorithm making mistakes, their attitude can quickly change from appreciation to aversion [11]. And with regard to the third point, if the algorithm is presented as more capable than a human agent, appreciation is more likely than aversion [10]. As noted earlier, another key factor to consider is whether the task or decision at hand involves high or low stakes, as this might influence the basic human tendency for algorithmic appreciation or aversion. This difference between high and low stakes might be additionally modulated by whether humans are more or less risk averse. Further factors include the prior description of an algorithm (see recent review [13]) and various kinds of affective influences such as whether a robot has human facial features (like *iCub*) or not. Finally, it matters whether a person is asked to decide if she would rather perform a task herself or to (partially) delegate it to an algorithm, or whether she is asked to decide if a task should be performed by *another* human or an algorithm.

To date, however, research has not investigated to what extent humans show offloading in tasks with high attentional demands. Are humans willing to (partially) offload task demands to an algorithm in order to reduce their own attentional load? This question is highly relevant since humans tend to commit more errors under high attentional load [14] and could thus substantially profit from cognitive offloading. By sharing a task with an algorithm, humans could not only reduce cognitive effort and thereby increase individual accuracy but could also go beyond their own capacity limitations [14] and thus achieve an overall performance level that exceeds the level they would achieve if they performed the entire task alone.

Thus, in the present study, we investigated whether and to what extent humans offload parts of an attentionally demanding task to an algorithm–and whether prior information about this algorithm affects the human tendency for offloading.

We addressed this question by conducting two behavioral experiments in which participants performed a multiple object tracking (MOT) task [15]. We chose this task because it has been reliably used in earlier research to test the limitations of attentional processing [16] and to investigate human-human [17] as well as human-AI collaborations [18, 19]. In the MOT task, participants are required to track a subset of moving target objects among distractor

objects on a computer screen. Studies showed that humans are able to track a limited number of objects (four objects [20]; but also see: [21]) and that tracking is an effortful task that requires sustained attention over prolonged periods of time (e.g., [22, 23]). Thus, the MOT task is highly suitable to tax the human attentional system and therefore provides an ideal "test case" for our purposes.

With regard to the outcome of the present study, two opposite hypotheses can be made based on prior research. On the one hand, earlier findings suggest that humans will not offload tasks to an algorithm that acts autonomously, as they prefer algorithms that have an advisory function only [4]. On the other hand, given that the present task is attentionally demanding, one could predict that humans will (partially) offload this task to an algorithm to reduce their own cognitive effort [1]–in line with the offloading tendencies humans show nowadays in certain daily activities (e.g., when using *Google Maps* or *Siri*). Given that the present study uses a scenario with what can be considered rather low stakes, the second hypothesis seems more likely, as human offloading tendencies may arguably be stronger when low stakes are involved.

Moreover, as previous research demonstrated that the human tendency for algorithmic appreciation vs. aversion may depend on the (ascribed) reliability [5] and expertise of an algorithm [7, 8, 10], we explored whether including the expertise factor would affect participants' behavior also in the present study. For this reason, we conducted one experiment in which participants did not receive any prior information about the expertise of the algorithm (Experiment 1; "computer capacity unknown") and a second experiment in which participants were informed beforehand that the algorithm was flawless (Experiment 2; "computer capacity known").

## Materials and methods

### Participants

Fifty-two university students participated in the present study, i.e., 26 participants took part in Experiment 1 ($M$ = 25.82 years, $SD$ = 4.30 years, 11 female, 15 male) and Experiment 2 ($M$ = 26.85 years, $SD$ = 8.46 years, 18 female, 8 male), respectively. Sample size was determined using G*Power [24] (alpha = 0.05, power = 0.80) with the aim to detect medium-sized effects both for correlations ($r$ = 0.5) and pairwise comparisons (Cohen's $d$ = 0.58). Participants gave written informed consent prior to participation and received 10 EUR as compensation. All participants expressed their consent for publication. The study was conducted in line with the ethical principles of the Declaration of Helsinki and of the American Psychological Association (APA). The study was approved by the ethics committee of the Institute of Philosophy and Educational Research at Ruhr University Bochum (EPE-2023–003).

### Experimental setup and procedure

Participants sat at a desk in front of a 24" computer screen (resolution: 1920 x 1080 pixels, refresh rate: 60 Hz), at a distance of 90 cm. A keyboard and mouse were positioned within easy reach.

Experiment 1 consisted of two conditions, a Solo condition (always performed first) and a Joint condition (always performed second). We chose this fixed order of conditions because we wanted to see whether (and to what extent) participants would reduce their maximum individual tracking load in the Joint condition by offloading to the computer partner. For participants to decide how much to offload, they needed to be aware of their own individual capacity. This means that in case of the MOT task, participants needed to know how many targets they were able to track individually. For this reason, we had participants perform the Solo condition first (so they could learn about their individual capacity), followed by the Joint condition

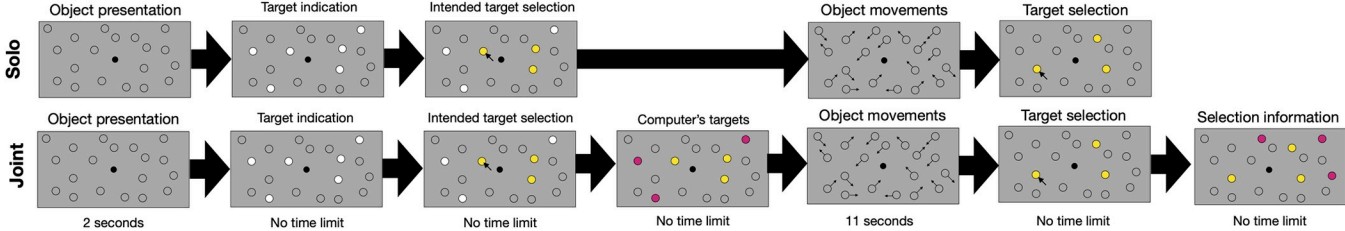

**Fig 1.** Top row: Exemplary trial sequence for the Solo condition. Bottom row: Exemplary trial sequence for the Joint condition.

where participants could then decide whether they wanted to reduce their tracking load relative to the Solo condition. Note that we address potential concerns about order effects in the final paragraph of the Results section.

There were 25 trials in the Solo and 50 trials in the Joint condition; resulting in 75 trials in total. In the Solo condition, participants performed the MOT task alone. In each trial, 19 stationary objects were initially displayed on the screen. Out of these, 6 objects were randomly selected as "targets" and were highlighted in white; all other objects served as "distractors" and were colored in grey. Participants were instructed to select as many (i.e., between 0 and 6) targets as they would like to track in that trial. They indicated their selection via mouse click and then confirmed this selection by clicking on a dot in the center of the screen. Upon confirmation, the highlighted targets switched color such that all objects (targets and distractors) now looked identical, i.e., all were colored grey. All objects then started to move, moving across the screen in randomly selected directions for a duration of 11 seconds. While moving, objects repelled each other and the screen borders in a physically plausible way (i.e., angle of incidence equals angle of reflection). Once the objects stopped moving, participants were instructed to click on those targets they had previously selected. They confirmed their target selection by clicking on the dot in the center of the screen, thereby marking the end of the trial. For an exemplary trial sequence, see Fig 1 (top row).

Participants were informed that they would earn one point for each correctly identified target, yet would lose one point for each incorrect selection. The goal was to identify as many targets as possible without making mistakes. Prior to performing the Solo condition, participants were instructed by the experimenter on the task procedure and performed two training trials to become familiar with the sequence of events.

In the Joint condition, participants performed the same MOT task as in the Solo condition but now together with a so-called "computer partner" (which was an algorithm). Participants were told that those targets that they do not select themselves will be selected and tracked by the computer. That is, after participants selected their targets and confirmed their selection, the remaining targets were highlighted in violet to indicate that these had been selected by the computer partner. For instance, if a participant selected three particular targets, the remaining three targets were automatically selected by the computer partner. Then, as in the Solo condition, all objects started moving across the screen (with targets and distractors looking identical) for 11 seconds. Once objects stopped moving, participants were asked to select their previously selected targets. After participants confirmed their selection, the remaining targets were selected by the computer partner; the selected targets were highlighted in yellow. Participants then confirmed seeing the computer partner's selection by pressing the space bar, thereby marking the end of the trial. For an exemplary trial sequence, see Fig 1 (bottom row).

As in the Solo condition, participants were informed that they would earn one point for each correctly identified target and lose one point for each incorrect selection, and that the same was true for the targets selected by the computer partner. Participants were told that the

total score consisted of the sum of their own and the partner's points. The goal was to identify, jointly with the partner, as many targets as possible without making mistakes. The tracking accuracy of the computer partner was 100% (i.e., its selections were always correct), yet participants were not explicitly informed about this. Prior to performing the Joint condition, participants were instructed by the experimenter about the task procedure and performed two training trials.

After completing both conditions, participants were presented with several task-related questions. First, they were asked how many targets they had tracked (and why) and whether they had followed a certain strategy when selecting which targets to track. These questions were asked separately for each condition. In addition, for the Joint condition, participants were asked how they decided how many targets to track themselves and how many targets to "offload" to the computer partner (note that the word "offload" was not explicitly used). Lastly, participants were asked to rate how many targets they believed the computer partner was able to track (on a scale from "0 targets" to "more than 6 targets").

After answering the questions above, participants were presented with a set of questionnaire items that capture personality traits. This set was designed such that participants' replies might provide insight into the factors that influence a person's willingness to offload tasks to an algorithm. The set consisted of the Desirability of Control Scale (20 items [25]), the Trust in Automation Scale (3-items-subset [26]), and the Affinity for Technological Systems Scale (9 items [27]). In addition, participants were asked to rate how reliable and competent they perceived the computer partner to be (7 items). This last set of items was also taken from the Trust in Automation Scale [26]) but was slightly adapted in wording such that it referred specifically to the computer partner. The exact wording of all items (in German), as well as links to the English translation, are provided in S1 File.

In Experiment 2, the procedure was exactly the same as in Experiment 1, with only one difference: Prior to performing the Joint condition with the computer partner, participants were informed that the computer partner is a software specifically designed for the MOT task such that it tracks targets with 100% accuracy regardless of how many targets it is assigned to track (see S1 File for the exact wording in German). In other words, they were told that the computer partner performs the task flawlessly, with an unlimited tracking capacity. Note that the tracking accuracy of the computer partner was 100% in both Experiments 1 and 2; the only difference was that in Experiment 2, participants were made aware of this fact whereas participants in Experiment 1 were uninformed.

The experiments were programmed in Python 3.0 using the *pygame* library. An experimental session (i.e., completing the experiment as well as the questionnaires) took about 60 minutes. All analyses were performed using custom R scripts.

## Results

In Experiment 1, to address the question of whether humans offload (parts of) an attentionally demanding task to an algorithm, we compared how many targets participants chose to track in the Solo condition compared to the Joint condition (for a descriptive overview, see Fig 2A). We found that participants chose to track significantly fewer targets in the Joint ($M = 2.68$, $SD = 1.08$) compared to the Solo ($M = 3.42$, $SD = 0.55$) condition, paired t-test: $t(25) = -3.67$, $p = .001$, Cohen's $d = 0.72$ (medium-sized effect [28]). This indicates that, in the Joint condition, participants decided to offload a subset of targets to the computer partner. Note that in the Solo condition, participants tracked between 3 and 4 targets on average, indicating that they tracked (close to) the maximum number of four targets humans are typically capable of tracking [21].

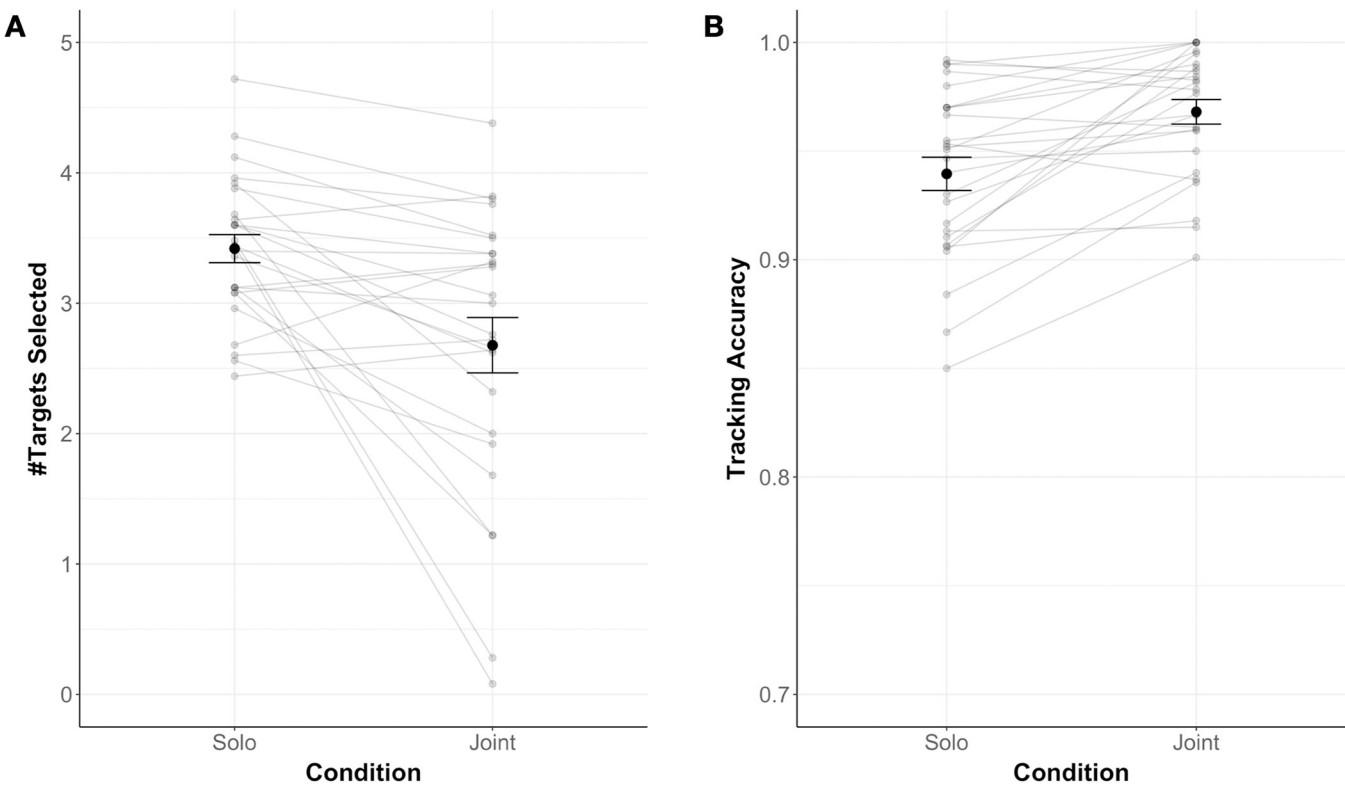

**Fig 2. Data overview for Experiment 1.** The average number of targets participants selected (A; #Targets Selected) and the average accuracy with which participants tracked these targets (B; Tracking Accuracy) are displayed as a function of condition (Solo vs. Joint). Note that for both conditions, only the participants' accuracy is shown. Errors bars show Standard Error of the Mean.

We next explored how offloading affected participants' tracking accuracy (for a descriptive overview, see Fig 2B). Accuracy was computed by dividing participants' correct selections by all selections for each trial. We found that accuracy was significantly higher in the Joint ($M = 0.97$, $SD = 0.03$) compared to the Solo ($M = 0.94$, $SD = 0.04$) condition (paired t-test: $t(25) = 4.59$, $p < .001$, Cohen's $d = 0.90$ (large-sized effect [28]), suggesting that offloading targets to the computer partner increased the accuracy of participants' individual performance.

For Experiment 2, we first performed a belief manipulation check, i.e., we verified whether participants had actually believed that the computer partner was flawless (i.e., tracked with 100% accuracy), in line with what they had been told by the experimenter. For this purpose, we examined participants' replies to the question "How many targets can the computer partner track accurately?" (see S1 File). Note that this question was asked *after* participants had performed the Joint condition. We found that the computer's capacity was rated as significantly higher in Experiment 2 compared to Experiment 1 ($\chi^2(5) = 16.83$, $p = .005$), see Fig 3. Twenty-five out of the twenty-six participants rated the computer's capacity to be "6 targets" (5 participants) or higher (20 participants). This confirms that most participants in Experiment 2 actually believed that the computer partner could flawlessly track at least 6 targets (i.e., the maximum number of targets in the present task).

Consistent with the analyses performed for Experiment 1, we analyzed the number of targets participants selected and participants' tracking accuracy also for Experiment 2. Results showed that participants selected significantly fewer targets (paired t-test: $t(25) = -6.60$, $p < .001$, Cohen's $d = 1.29$ (large-sized effect [28]) and performed with a significantly higher

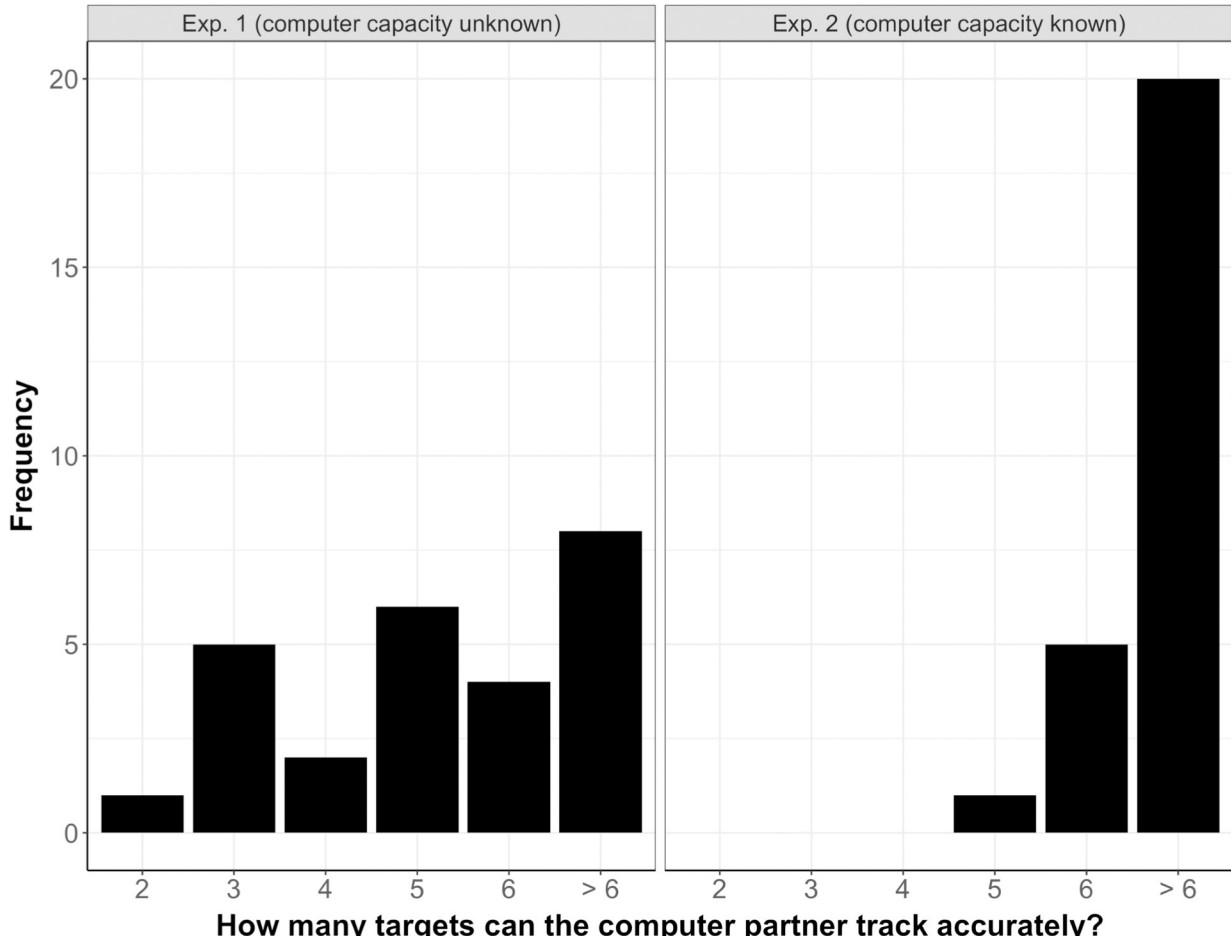

**Fig 3.** Frequency of rated capacity for Experiment 1 (left panel) and Experiment 2 (right panel).

accuracy (paired t-test: $t(24) = 4.08$, $p < .001$, Cohen's $d = 0.81$ (large-sized effect [28]) in the Joint ($M_{targets} = 2.19$, $SD_{targets} = 1.06$; $M_{accuracy} = 0.97$, $SD_{accuracy} = 0.04$) compared to the Solo ($M_{targets} = 3.35$, $SD_{targets} = 0.56$; $M_{accuracy} = 0.93$, $SD_{accuracy} = 0.06$) condition (for a descriptive overview, see Fig 4). This is in line with the findings from Experiment 1 and indicates that, in the Joint condition, participants offloaded a subset of targets to the computer partner. This offloading was accompanied by a boost in individual accuracy.

To assess whether the extent of offloading was larger in Experiment 2 than in Experiment 1 (i.e., whether the prior information about the computer's capacity influenced participants' willingness to offload), we compared the number of targets selected in Experiments 1 versus 2. There was no significant difference, independent t-test: $t(50) = 1.59$, $p = .117$, Cohen's $d = 0.44$ (small-sized effect [28]). When calculating a Bayes factor for this comparison, we found that the null hypothesis is 1.27 more likely than the alternative hypothesis.

Turning to the questionnaire results, we first analyzed participants' ratings regarding the number of targets they had tracked in the Solo and Joint condition (for a descriptive overview, see Fig 5). Participants' ratings were consistent with their actual behavior, i.e., participants stated that they had tracked fewer targets in the Joint vs. Solo condition both for Experiment 1 (paired t-test: $t(25) = 3.58$, $p = .001$, Cohen's $d = 0.70$ (medium-sized effect [28])) and Experiment 2 (paired t-test: $t(25) = 4.97$, $p < .001$, Cohen's $d = 0.98$ (large-sized effect [28])). This

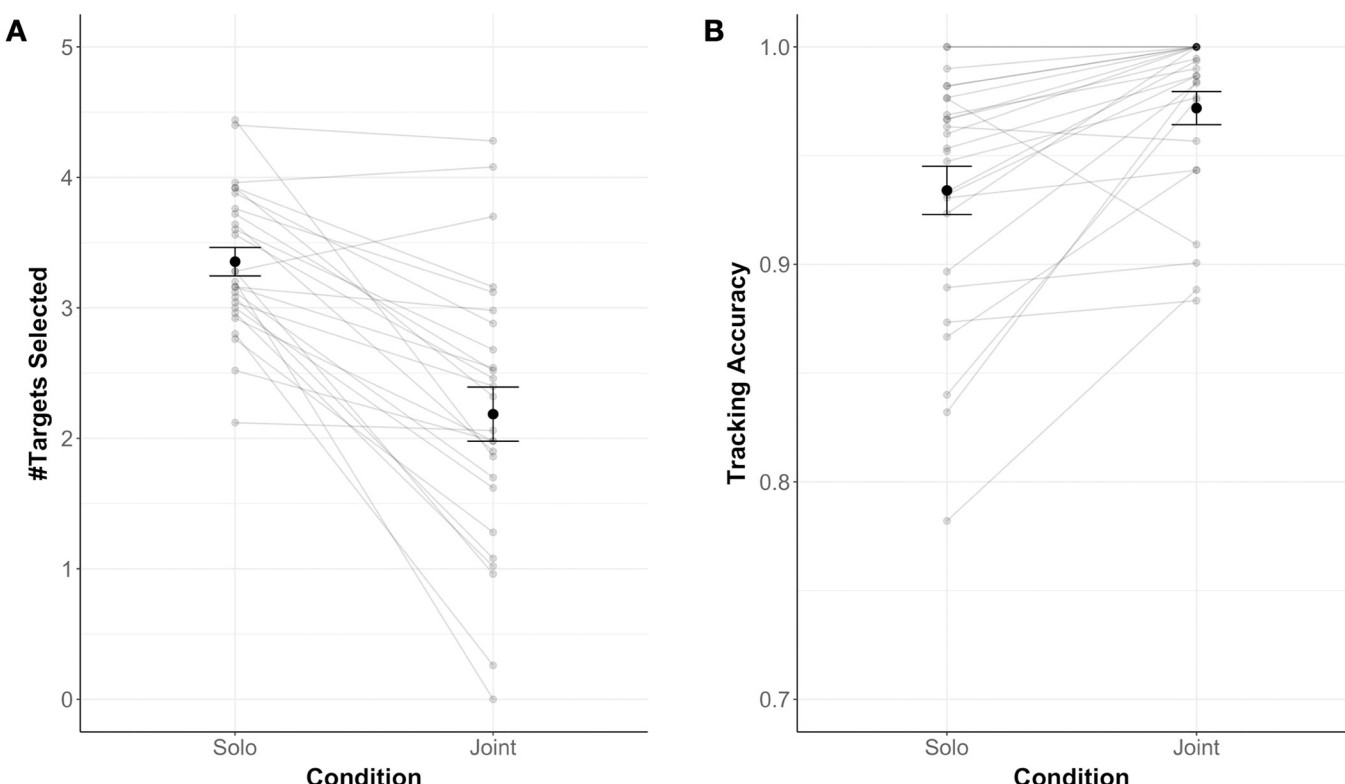

**Fig 4. Data overview for Experiment 2.** The average number of targets participants selected (A; #Targets Selected) and the average accuracy with which participants tracked these targets (B; Tracking Accuracy) are displayed as a function of condition (Solo vs. Joint). Errors bars show Standard Error of the Mean.

difference is not significant across experiments (independent t-test: $t(50) = -0.37$, $p = .712$, Cohen's $d = 0.10$ (negligible effect [28])).

We next categorized the strategies that participants described when asked about how they had selected which targets to track. In the Solo condition, by far the most prevalent strategy across experiments was to choose objects that were initially displayed in close proximity on the screen. The majority of participants (85% in Exp. 1; 77% in Exp. 2) reported this strategy; the rest did not indicate any clear strategy.

In the Joint condition, we had asked participants how they decided how many targets to track themselves and how many targets to "offload" to the computer partner. To evaluate participants' strategies, we created the following four categories based on participants' responses: (1) *No change*: Participants did not change their behavior relative to the Solo condition; (2) *Fair*: Participants preferred a fair split (selecting 3 targets for themselves and offloading 3 targets to the computer partner); (3) *Minor offloading*: Participants offloaded targets to a small extent (selected 0.5–1 target fewer in the Joint compared to the Solo condition); (4) *Major offloading*: Participants offloaded targets to a large extent (selected 1.5–3.5 targets fewer in the Joint compared to the Solo condition). Note that, if participants indicated a range (e.g., 1–2 targets), we computed the average of the bounds of the range (e.g., 1.5 targets). When comparing the frequency of the reported strategies in the Joint condition across experiments (for a descriptive overview, see Fig 6), we observed that around 40% of participants (i.e., 11 out of 26 participants) indicated no change in behavior relative to the Solo condition in Experiment 1 whereas in Experiment 2, the same number of participants (i.e., 11 out of 26 participants) reported minor offloading. However, this difference was not significant ($\chi^2(3) = 3.17$, $p = .365$). Focusing only on the frequency of participants indicating offloading (either minor or

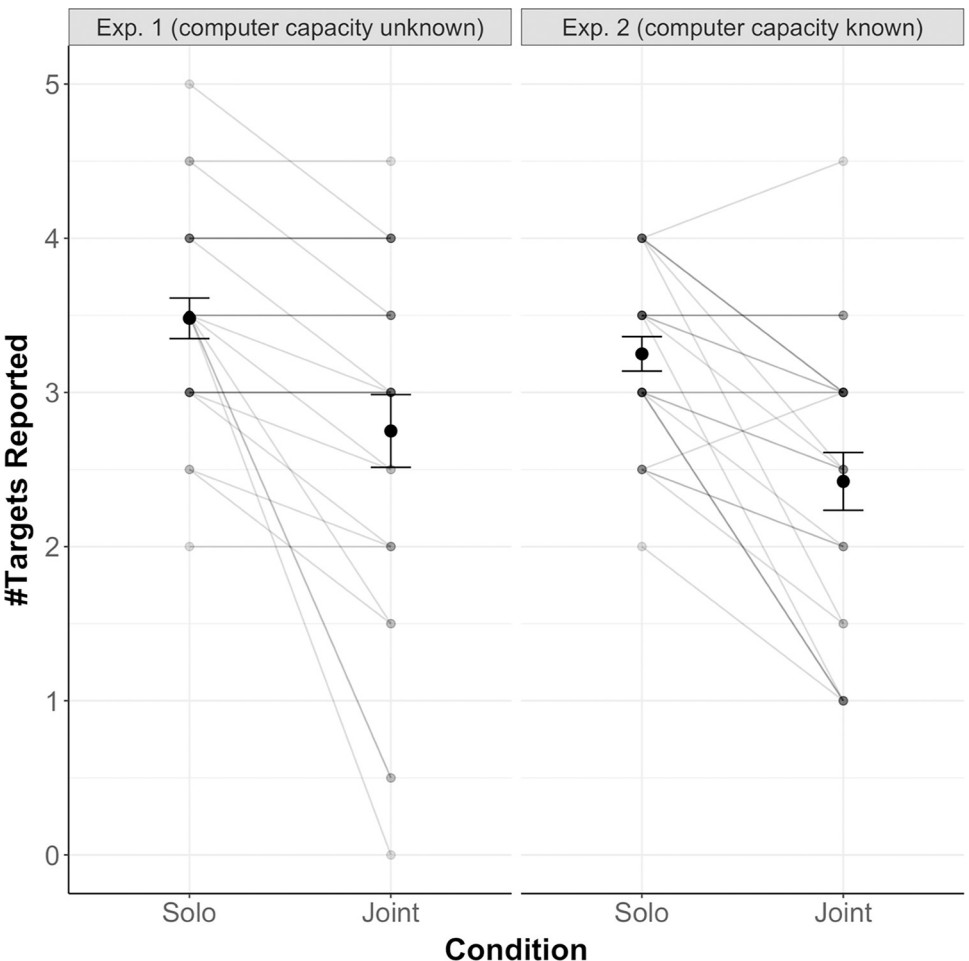

**Fig 5. The number of targets participants reported to have tracked, displayed as a function of condition (Solo vs. Joint) and experiment (left panel: Exp. 1, right panel: Exp. 2).** Error bars show Standard Error of the Mean.

major), we observed that 11 participants (42%) in Experiment 1 and 17 participants (65%) in Experiment 2 reported offloading as a strategy in the Joint condition.

We next tested to what extent our questionnaire scales predicted offloading. For this purpose, we combined the data from our two experiments and first evaluated the reliability of all our questionnaire scales using Cronbach's alpha [29]. We found acceptable reliabilities for all our scales (Desirability of Control: 0.78; Affinity for Technological Systems: 0.91; Trust in Automation: 0.69; Reliability and Competence of the Computer Partner: 0.73) [30]. We entered all questionnaire scales as metric predictors and the between-subjects factor Experiment (1 vs. 2) into a multiple regression and used the offloading extent (i.e., difference between number of selected targets in the Solo vs. Joint condition) as the dependent variable ($F(5,46) = 1.00$, $p = .426$, $R^2 = 0.10$). None of the predictors was significant (Desirability of Control: $\beta = -0.06$, $t = -0.42$, $p = .679$; Affinity for Technological Systems: $\beta = 0.07$, $t = 0.450$, $p = .655$; Trust in Automation: $\beta = -0.12$, $t = -0.81$, $p = .422$; Reliability and Competence of the Computer Partner: $\beta = -0.18$, $t = -1.26$, $p = .213$), suggesting that none of the questionnaire scales predicted how many targets participants decided to offload.

Finally, we addressed a potential confound inherent in our experimental design: Because participants completed the two conditions always in a fixed order, with the Joint after the Solo

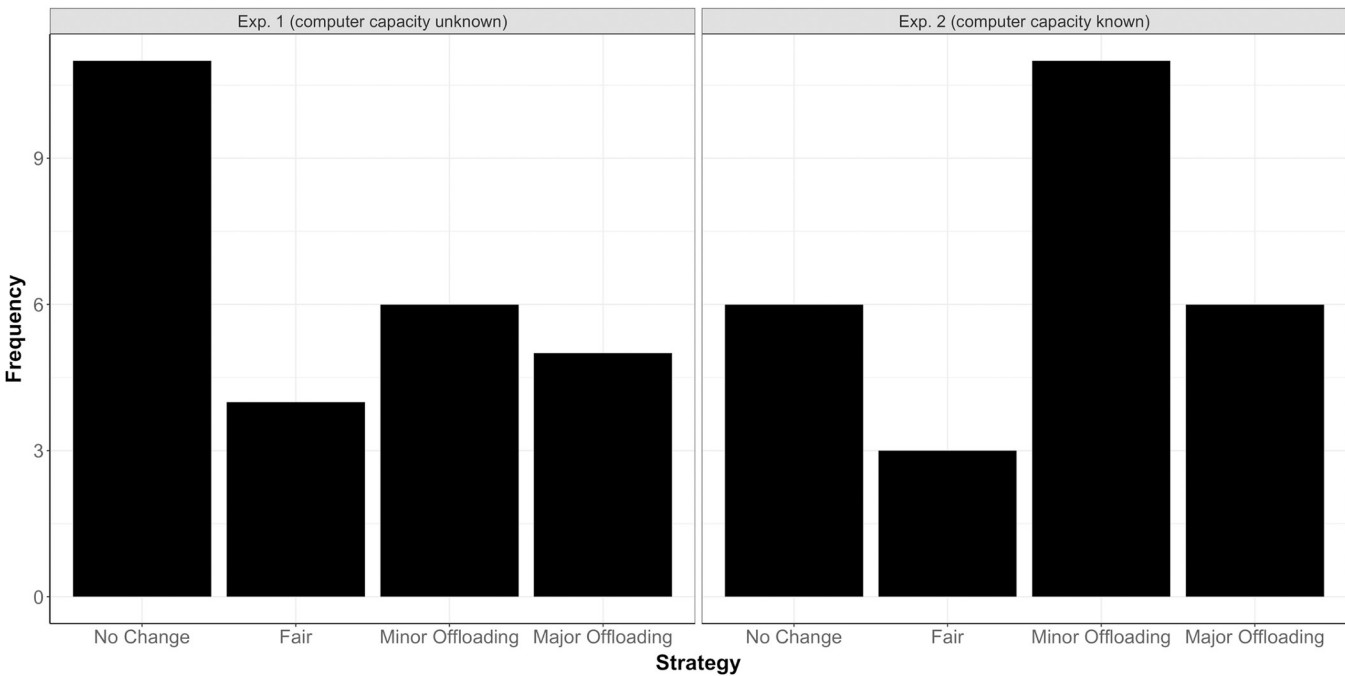

**Fig 6. Frequency of strategies in the Joint condition, shown separately for Experiment 1 (left panel) and Experiment 2 (right panel).**

condition, it is possible that the difference we observed between the two conditions can be simply ascribed to effects of fatigue. In other words, maybe participants decided to track fewer targets in the Joint compared to the Solo condition simply because they felt tired–and not because they wanted to offload parts of the attentional demands to the computer partner. If this was true, the same drop in selected targets should occur if the first Solo condition was followed by a second Solo (rather than Joint) condition, because participants should be equally fatigued over time. We addressed this possibility by drawing on data from an earlier study conducted by the first author ([17]; see "No Information" condition). In that study, 32 participants performed the same MOT task as in the present study for 100 trials; the present study consisted of 75 trials in total. Note that participants in that study took part in pairs of two yet performed the task individually, just as in the Solo condition of the present study. We split participants' data up in 25-trial segments and tested whether, over time, the number of objects participants selected and their tracking accuracy decreased. For this purpose, we used a linear mixed model with random intercepts for each pair and included the within-subjects factor Trial Segments (1st Quarter, 2nd Quarter, 3rd Quarter, 4th Quarter), for which we defined the 1st Quarter as the reference group (i.e., all other trial segments are compared to this reference group). We found that the number of selected targets did not significantly differ relative to the reference group (1st Quarter vs. 2nd Quarter: $t(122) = -0.71$, $p = .479$; 1st Quarter vs. 3rd Quarter: $t(122) = -0.60$, $p = .551$; 1st Quarter vs. 4th Quarter: $t(122) = -0.10$, $p = .918$). The same was true for participants' tracking accuracy (1st Quarter vs. 2nd Quarter: $t(123) = 0.52$, $p = .602$; 1st Quarter vs. 3rd Quarter: $t(122) = 0.51$, $p = .612$; 1st Quarter vs. 4th Quarter: $t(122) = 0.11$, $p = .915$). In sum, these results suggest that, in a MOT task with up to 100 trials, neither the number of selected targets nor the tracking accuracy decreases over time as a result of fatigue. Accordingly, the observed difference between the Solo and Joint condition in the present study can hardly be explained by effects of fatigue.

## Discussion

In the present study, we investigated whether and to what extent humans offload parts of an attentionally demanding task to an algorithm–and whether prior information about this algorithm affects the human tendency for offloading. In two experiments, participants performed a multiple object tracking (MOT) task, either alone (Solo condition) or together with a so-called computer partner (Joint condition). Across experiments, participants showed a clear willingness to offload parts of the task (i.e., a subset of to-be-tracked targets) to the computer partner. In particular, participants decided to track 3.4 targets on average in the Solo compared to 2.4 targets in the Joint condition, i.e., they offloaded 1 target to the computer partner. Earlier research suggests that this difference in the number of tracked targets cannot be alternatively explained by a progressive performance decrease over time due to fatigue. This result is in line with earlier research predicting that humans will (partially) offload an attentionally demanding task to an algorithm to reduce their own cognitive effort [1]. Participants' behavior in the present study is also consistent with the offloading tendencies humans show nowadays in certain daily activities (e.g., when using *Google Maps* or *Siri*).

The decision to offload a subset of targets to the computer partner resulted in a boost in participants' tracking accuracy, indicating that participants made fewer mistakes thanks to a reduction in cognitive demand [1]. This result demonstrates that offloading benefited participants' individual performance.

As previous research demonstrated that the human tendency for algorithmic appreciation vs. aversion may depend on the (ascribed) reliability [5] and expertise of an algorithm [7, 8, 10], we also explored whether the expertise factor would affect participants' behavior in the present study. To this end, participants in Experiment 1 did not receive any prior information about the expertise of the algorithm whereas participants in Experiment 2 were informed beforehand that the algorithm was flawless. Results showed that this prior information did not significantly affect the number of targets participants decided to track.

Regarding our main result, i.e., the observation that participants decided to track fewer targets in the Joint compared to the Individual condition, one may wonder whether this difference could be due to the fact that in the Joint condition, participants might have tried to monitor the computer's performance *in addition* to performing their own tracking task. If this was true, then this might have impaired participants' individual tracking capacity, resulting in a smaller number of tracked targets in the Joint condition. Note that this worry concerns Experiment 1 only, as in Experiment 2, participants were told that the computer's accuracy was perfect such that there was no obvious need for monitoring. From participants' questionnaire responses, we know that indeed, in Experiment 1, several (i.e., 19%) participants tried to find out about the computer's accuracy through monitoring the targets selected by the computer. However, participants quickly realized that the computer's accuracy was high and stopped monitoring after only a few trials. No participant reported having monitored the computer throughout. These subjective self-reports are supported by participants' objective accuracy scores: if participants had monitored the computer throughout the entire experiment, then their own performance should have been severely impaired. Instead, participants' performed with the same average accuracy of 97% in Experiment 1 and 2, while the number of tracked targets was similar. The fact that accuracy levels did not differ between experiments suggests that participants did not perform a secondary task in Experiment 1.

Moreover, our results show that participants were well aware of their tracking/offloading decisions. When asked after the experiment how many targets they had tracked in the Solo and Joint condition, the reported numbers were consistent with the numbers they had actually tracked. This suggests that participants had a high metacognitive awareness of their actions

and that the offloading was likely an explicit choice participants made, which fits well with the proposition that metacognitive evaluations play a central role in offloading [1].

Finally, our questionnaire results do not provide any support for the hypothesis that participants' offloading tendencies might be related to certain personality dimensions such as desirability of control [25], trust in automation [26], and affinity for technological systems [27]. However, future studies with larger sample sizes are needed to rule out any influence of these personality dimensions more decisively. Alternatively, future studies could test these factors experimentally, e.g., by comparing two groups of pre-selected participants (e.g., one group with a high and one with a low affinity for technological systems).

We suggest that future studies could consider the degree of cognitive load also as a moderating variable. For instance, for medical decisions, people may be more inclined to rely on the advice of an algorithm rather than a human doctor if they know that the human doctor formulated her advice under high cognitive load (e.g., at a particular busy time in the hospital). This is because people might realize that the performance of the algorithm, in contrast to that of the human, cannot be affected by any inherently human cognitive capacity limitations. Note that, as mentioned at the outset, future studies would also need to investigate the effect of cognitive load in high stakes scenarios. As the experimental task used in the present study involves considerably low stakes, our results cannot be generalized to high stakes scenarios such as the afore-mentioned medical decision-making. Investigating the latter types of scenarios seems worthwhile because it is in these scenarios that errors committed by humans under high attentional load [14] can have serious consequences, and thus, humans could substantially profit from cognitive offloading. Hence, understanding the offloading tendencies of humans (e.g., doctors) in such high stakes scenarios is critical.

More generally, the conclusions drawn from the present study are limited in the sense that we have only investigated the willingness of human participants to offload a task to a computer partner, yet we did not test their willingness to offload the same task to a human partner. Thus, based on the present findings, we can report that humans are willing to (partially) offload an attentionally demanding task to an algorithm, but we do not know to what extent they would offload this task to another human, nor whether, if given the choice, they would choose the computer or the human as a partner. These questions need to be addressed by future studies to gain a more comprehensive picture of the human aversion/appreciation tendencies.

A final open question one must ask in light of the results of the present study is why, in Experiment 2, participants did not decide to offload the *complete* task to the computer partner–given that they knew that the computer would perform the task flawlessly. Two evident explanations for why participants did not offload the entire task are the following. (1) Participants did not want to feel bored while passively waiting for the computer to finish the task, and (2) participants experienced some form of experimental demand, i.e., they wanted to continue performing the experimental task that they had been asked (and paid) to perform. To address these points, we are currently conducting a follow-up study in which we inform participants that, if they were to offload the tracking task entirely to the computer partner, then they would be able to perform a secondary task while the computer performs the tracking task. This way, participants should feel free to offload the entire task–they would not feel bored nor would they have the impression not to meet the experimenter's demands. Likely, the specifics of this secondary task will influence participants' behavior, e.g., it will matter whether the secondary task is easier or more difficult than the tracking task, whether it is incentivized (e.g., by an additional monetary gain), and whether it allows participants to still monitor the computer's performance.

To conclude, the present findings show that people are willing to (partially) offload task demands to an algorithm to reduce their own cognitive effort, thereby increasing individual

accuracy. We suggest that the cognitive load of a task is an important factor to consider when evaluating human tendencies for offloading cognition onto artificial systems.

## Supporting information

**S1 File. Prior instruction for Experiment 2 & questionnaires.**
(PDF)

## Author Contributions

**Conceptualization:** Basil Wahn, Laura Schmitz.

**Data curation:** Frauke Nora Gerster.

**Formal analysis:** Basil Wahn.

**Funding acquisition:** Matthias Weiss.

**Investigation:** Basil Wahn, Frauke Nora Gerster.

**Methodology:** Basil Wahn.

**Project administration:** Basil Wahn.

**Software:** Basil Wahn.

**Supervision:** Matthias Weiss.

**Visualization:** Basil Wahn.

**Writing – original draft:** Basil Wahn, Laura Schmitz.

**Writing – review & editing:** Basil Wahn, Laura Schmitz, Frauke Nora Gerster, Matthias Weiss.

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
