## [Decision Letter · Decision Letter 0]

6 Mar 2023

PONE-D-23-00394

Algorithmic appreciation under cognitive load: Humans are willing to offload parts of an attentionally demanding task to an algorithm

PLOS ONE

Dear Dr. Wahn,

Thank you for submitting your manuscript to PLOS ONE. After careful consideration, we feel that it has merit but does not fully meet PLOS ONE’s publication criteria as it currently stands. Therefore, we invite you to submit a revised version of the manuscript that addresses the points raised during the review process.

We look forward to receiving your revised manuscript.

Kind regards,

Jan C. Zoellick

Academic Editor

PLOS ONE

Journal Requirements:

  "This study is part of the cooperative research project "INTERACT!", which received funding from the program "Profilbildung 2020" — an initiative of the Ministry of Culture and Science of the State of North Rhine-Westphalia. The sole responsibility for the content of this publication lies with the authors."

Additional Editor Comments:

Both reviewers agree that the manuscript will provide a valuable contribution to the discourse when revised. When revising the manuscript make sure to answer to all comments and include even the minor points. I would like to draw particular attention to the following points:

- Interpretation of results: Reviewers 1 and 2 comment on several issues regarding the interpretation of results. With the task on hand being of relatively low valence, does the experiment really measure algorithmic aversion? Without comparison to a human partner, does the experiment really measure algorithmic aversion? In your revision, please make sure to reflect on these issues and draw on additional examples from the literature regarding low valence tasks. Also focus on cognitive load vs. maximum cognitive capacity when comparing results from experiments 1 and 2.

- Trustworthiness of results: Please also make sure to reflect on comparisons between experiments 1 and 2 as there are some concerns that they measure something completely separate. Be wary of the type 1 errors - with sometimes barely significant results (p = .044) and different experimental settings your study is not replicated yet. Without replicating results this might have been a random finding. Please double-check to not introduce false positive findings into the literature.

- Accuracy of reporting: Both reviewers find several instances where signs seem to be opposite than expected or you describe majorities when minorities were more suitable. Please double-check the results and interpretation section carefully and ensure to accurately describe all rersults (also regarding description of references).

- Open data: Please provide the dataset and analysis syntax open without restrictions to comply with PLOS ONE publication criteria.

Reviewers' comments:

Reviewer's Responses to Questions

**Comments to the Author**

1. Is the manuscript technically sound, and do the data support the conclusions?

Reviewer #1: Partly

Reviewer #2: Partly

2. Has the statistical analysis been performed appropriately and rigorously? 

Reviewer #1: Yes

Reviewer #2: Yes

3. Have the authors made all data underlying the findings in their manuscript fully available?

Reviewer #1: Yes

Reviewer #2: Yes

4. Is the manuscript presented in an intelligible fashion and written in standard English?

Reviewer #1: Yes

Reviewer #2: Yes

5. Review Comments to the Author

Reviewer #1: Based on previous work showing variability in algorithmic aversion – where despite widespread use of smartphone apps (such as Siri or Google Maps), individuals still do not trust an algorithm to make medical assessments or forecasting as much as a human, or at least in an unsupervised way. The current work tests to see whether cognitive load can influence algorithmic aversion. The Authors use a multiple object tracking (MOT) task which requires participants to choose how many targets they would like to track and then after the objects move, they are asked to select the original targets. By first having participants perform the task unassisted, they develop a baseline of participant behavior and allow the participants to evaluate their own abilities/working memory capacity. Subsequently, participants are given the opportunity to let a computer partner track some of the objects and offload the work of the task to the algorithm. In Experiment 1, participants are not told how accurate the computer partner is, but in Experiment 2, participants are explicitly told that the algorithm has a 100% success rate. By comparing the relative willingness to offload work to the computer based on accuracy (i.e., Experiments 1 vs. 2), the current studies test the hypothesis that individuals are more likely to let a computer partner do some of the work if they believe it to perform well.

The Authors conclude that individuals will partially offload task demands to a computer if the task exceeds their own cognitive capacity. The evidence for this comes primarily from the “how many objects tracked” results of participants in the Solo vs. Joint conditions: where across both Experiment 1 and 2, participants chose to track fewer objects when the computer partner was able to assist relative to when they were doing so themselves. They also conclude that in certain conditions, individuals will be more likely to offload work if the computer partner is assumed to be more accurate. This effect wasn’t supported by a direct comparison of Experiment 1 vs 2 but comes from the observation that when the data were split into sets of 10-trials, the number of objects chosen in the Joint task between Exp 1 and 2 were different in the first set of 10 trials. Additionally, several personality traits were examined that might affect participants’ willingness to offload work to the computer partner, though none were significantly related to this.

I think the article has several key strengths. In terms of motivating the current research question, the point that the “humans tend to commit more errors under high attentional load and could thus substantially profit from cognitive offloading” is a good one and motivates the current work nicely. The figures are great - Figure 1 is quite helpful for displaying the task design. I particularly like the way the data is presented in the plots in Figures 2-7. I appreciate that the researchers show the distribution of the data by plotting the individual data points! For the most part, the study is well designed, and the Authors have considered many of the potential limitations and pro-actively addressed them. I also think that with a couple exceptions, the conclusions are reasonable and well calibrated based on the results. Though not the primary goal of the study per se, I found it interesting that the participants didn’t trust the computer more in Exp 1. I have a few concerns about alternative explanations for this finding (see Major point #3 below) but an interesting result nonetheless.

For me, the major weaknesses relate to how this research is framed (Major points #1 & #4 below), the conclusions drawn from a very weak effect (Major point #2 below), and the fact that the data and code are not openly available (Major point #5).

My overall assessment is that this paper does show an interesting effect and the key results are sufficiently robust, so with some modifications it will be suitable for publication. I detail these major concerns and several minor points below.

Major Concerns:

1. Intro: I am not an expert in human-computer interactions so I’m not sure if there are any missing or inappropriate references for the background. However, some of the examples presented that are evidence for algorithmic aversion seem to be higher stakes than the current study (i.e., the example of a medical decision) or are not so much an accuracy decision, but an evaluation (the art authenticity example). The closest analog I see to the current study is that of using Google Maps or potentially the forecasting. It strikes me a bit as though the Authors are comparing apples and oranges a bit by lumping this all under the idea of algorithmic aversion/trust for humans over computers, when it might have something to do with risk aversion in the case of high stakes decisions. Perhaps there are more relevant, low stakes examples out there that could be drawn upon? At a minimum, including a discussion of other contextual factors that could influence algorithmic aversion (i.e., risk aversion, affective influences on decision making, whether the human is another person or oneself), would be helpful to situate the current study.

2. Results: The seemingly exploratory analyses comparing Exp 1 and 2 don’t appear to control for multiple comparisons at all. I suggest including MC correction for these results and removing the discussion of computer partner accuracy as a contributing factor, as the initial null result (comparing overall Exp 1 and Exp 2) and the weak effect for the first 10 trials (p = 0.044 uncorrected) doesn’t sugest a particularly robust or reliable effect.

3. Methods/Results: For the question of how many targets did participants think that the computer partner was able to track – I can see why the Authors did things in the order they did (i.e., question asked after all study procedures in Exp 1 and before starting joint condition in Exp 2). However, I wonder if this muddies whether conclusions can be drawn from comparing the number of items they assumed the computer could track between Experiments 1 and 2. That is, some participants in Exp 1 may have been making an absolute judgment (answering the question of “how accurate is the computer irrespective of what the participant did?”) and some might be reporting this based on what they saw: 6 minus the number that they usually chose. Maybe there is some additional evidence that I’m missing that mitigates this explanation. Or maybe there are some analyses that could be done to examine this (perhaps looking at the responses in Fig 3 as a function of number chosen)? Otherwise, I think it should be noted as a limitation.

4. Interpretation: There is some inconsistency in how the effects are described – as related to cognitive load or as exceeding cognitive capacity. For example, in the discussion, “The present findings show that humans are willing to (partially) offload task demands to an algorithm if these demands exceed their own attentional capacity, providing evidence of algorithmic appreciation (rather than aversion).” It seems to me that the data strongly support the idea that the participants did so to decrease their cognitive load, not because it exceeds their cognitive capacity (as in the Solo condition they were presumably at their capacity, and this is a decrease from that). I would recommend changing any discussion of capacity to effort/cognitive load based on this.

5. Data & code availability: The data are available for review but not openly available. In all my reviews, I request that the Authors upload their data and analysis code to a public repository to facilitate transparency and reproducibility of results.

Minor Concerns:

1. Results paragraph 1: “We found that participants chose to track significantly fewer targets in the Solo (M = 3.42, SD = 0.55) compared to the Joint (M = 2.68, SD = 1.08) condition (t(25) = -3.67, p = .001, Cohen’s d = -0.72), indicating that, in the Joint condition, participants decided to offload a subset of targets to the computer partner.” Are these condition means meant to be in the opposite order?

2. The experimental setup is described well and clearly. I think a bit more detail is needed on the analyses. It would be helpful to clearly state what the between and the within subjects effects were in any regressions. It would also help to know what software was used for the statistical analyses. I’m confused by the negative Cohen’s d values so I’m wondering if this is an oddity of a particular analysis software (or some R or python package/ library). These should be changed to positive values in any case.

3. It would be useful if the authors described the magnitude of the effects as well as I assume most folks don’t have the Cohen’s d scale memorized.

4. “When comparing the frequency of the reported strategies in the Joint condition across experiments (for a descriptive overview, see Figure 7), we observed that the majority (i.e., 11 out of 26 participants) indicated no change in behavior relative to the Solo condition in Experiment 1 whereas in Experiment 2, the majority (i.e., 11 out of 26 participants) reported minor offloading.” 11 out of 26 isn’t a majority of participants, do the Authors mean this was the most common response?

5. On page 17, I realize these are null results, but it would still be useful for the researchers to report beta values for the multiple regression in addition to or instead of the t-stats.

Other points:

As I do in all my reviews, I ran statcheck on this manuscript and found no inconsistencies.

Reviewer #2: Review of PONE-D-23-00394

Thank you for the opportunity to review this manuscript. It is an interesting read and a well-composed study with convincing analyses. There are some issues that raised my concern which may well be solved quickly.

If I understand correctly, in Joint, participants had another implicit task in Exp1 namely evaluating the performance of the partner to be able to decide how many objects to offload and thus to maximize the “points” they achieve together with the partner (this was set to 100% in Exp2 so no second task here). If in Exp1 Solo they chose as many objects as humanly possible (“in the Solo condition, participants tracked between 3 and 4 targets on average, indicating that they tracked (close to) the maximum number humans are typically capable of tracking [15]”), they would then not be able to keep up this high number of objects in Joint but they would have to reduce this number to achieve their additional second task, evaluating the partner. If this is the case, then Exp1 and Exp2 would be hard to compare. Also, considering this second implied task, participants did not really offload any objects in Exp1 Joint but in addition to their own objects also had to evaluate ALL other objects because they were selected automatically to be tracked by the partner. If this interpretation of mine is incorrect it needs to be clarified in the text so the reader does not make the same mistake.

I did not find any information on the actual accuracy of the partner (algorithm) in Exp1 but may have overlooked it. Assuming its accuracy is perfect, does not accuracy need to be higher in Joint because the partner automatically takes care of the objects not selected by the participant?

With regard to the general design of the study, is it really possible to conclude anything about algorithm appreciation or aversion? There is just this one type of partner (i.e., the algorithm) and no comparison against the same design with a human partner. It may well be that the participants would have behaved exactly the same when partnered with another human (in Exp1 evaluating their performance, in Exp2 trusting the claimed accuracy). This was my impression of some of the literature summarized in the introduction as well.

- Twice: “majority (i.e., 11 out of 26 participants)” needs to be minority?

- conference contributions better described as such instead of “Research Papers. 168.”

- In “We found that participants chose to track significantly fewer targets in the Solo (M = 3.42, SD = 0.55) compared to the Joint (M = 2.68, SD = 1.08) condition” here fewer needs to be more?

It’s a well-composed study and convincing to read with well-performed analyses, so I hope my questions can be answered quickly.

6. PLOS authors have the option to publish the peer review history of their article (what does this mean?). If published, this will include your full peer review and any attached files.

Reviewer #1: No

Reviewer #2: No

---

## [Author Response · Author response to Decision Letter 0]

14 Apr 2023

Please find our response letter in the uploaded "ResponseToReviewers_R1_PLOSONE_SUBMIT.docx" file.

---

## [Decision Letter · Decision Letter 1]

9 May 2023

Offloading under cognitive load: Humans are willing to offload parts of an attentionally demanding task to an algorithm

PONE-D-23-00394R1

Dear Dr. Wahn,

We’re pleased to inform you that your manuscript has been judged scientifically suitable for publication and will be formally accepted for publication once it meets all outstanding technical requirements.

Kind regards,

Jan Cornelius Zoellick

Academic Editor

PLOS ONE

Additional Editor Comments (optional):

Reviewers' comments:

Reviewer's Responses to Questions

**Comments to the Author**

1. If the authors have adequately addressed your comments raised in a previous round of review and you feel that this manuscript is now acceptable for publication, you may indicate that here to bypass the “Comments to the Author” section, enter your conflict of interest statement in the “Confidential to Editor” section, and submit your "Accept" recommendation.

Reviewer #1: All comments have been addressed

Reviewer #2: All comments have been addressed

2. Is the manuscript technically sound, and do the data support the conclusions?

Reviewer #1: Yes

Reviewer #2: Yes

3. Has the statistical analysis been performed appropriately and rigorously? 

Reviewer #1: No

Reviewer #2: Yes

4. Have the authors made all data underlying the findings in their manuscript fully available?

Reviewer #1: Yes

Reviewer #2: Yes

5. Is the manuscript presented in an intelligible fashion and written in standard English?

Reviewer #1: Yes

Reviewer #2: Yes

6. Review Comments to the Author

Reviewer #1: The Authors have responded to all of my inquiries and concerns very thoroughly. I appreciate their detailed response and think this manuscript is much improved. I recommend the revised manuscript for publication.

Reviewer #2: all comments have been answered and all concerns addressed. the paper will contribute to the literature.

7. PLOS authors have the option to publish the peer review history of their article (what does this mean?). If published, this will include your full peer review and any attached files.

Reviewer #1: **Yes: **Kimberly Meidenbauer

Reviewer #2: No

---

## [Editor Report · Acceptance letter]

12 May 2023

PONE-D-23-00394R1 

Offloading under cognitive load: Humans are willing to offload parts of an attentionally demanding task to an algorithm 

Dear Dr. Wahn:

I'm pleased to inform you that your manuscript has been deemed suitable for publication in PLOS ONE. Congratulations! Your manuscript is now with our production department. 

Kind regards, 

on behalf of

Dr. Jan Cornelius Zoellick 

Academic Editor

PLOS ONE